# The Effect of Post-Processing on the Mechanical Behavior of Ti6Al4V Manufactured by Electron Beam Powder Bed Fusion for General Aviation Primary Structural Applications

**Carmine Pirozzi** *,† [ID], **Stefania Franchitti** †, **Rosario Borrelli** † [ID], **Antonio Chiariello** and **Luigi Di Palma** [ID]

CIRA (Italian Aerospace Research Center), 81043 Capua CE, Italy; s.franchitti@cira.it (S.F.); r.borrelli@cira.it (R.B.); a.chiariello@cira.it (A.C.); L.DiPalma@cira.it (L.D.P.)

* Correspondence: c.pirozzi@cira.it; Tel.: +39-0823-3566

† These authors contributed equally to this work.

**Abstract:** In this work a mechanical characterization of Ti6Al4V processed by electron beam powder bed fusion additive manufacturing was carried out to investigate the viability of this technology for the manufacturing of flyable parts for general aviation aircraft. Tests were performed on different manufacturing conditions in order to investigate the effect of post processing as machining on the mechanical behavior. The study provides useful information to airframe designers and manufacturing specialists that work with this technology. The investigation confirms the low process variability and provides data to be used in the design loop of general aviation primary structural elements. The test results show a high level of repeatability indicating that the process is well controlled and reliable enough to match the airworthiness requirements. In addition, the so-called "as-built specimens", i.e., specimens produced by the electron beam melting machine without any major post-processing, have lower mechanical performances than specimens subjected to a machining phase after the electron beam melting process. Specific primary structural elements will be designed and flight cleared, resulting from the findings presented herein.

**Keywords:** additive manufacturing; EBM; mechanical characterization; general aviation; airworthiness

## 1. Introduction

The industrial need for multifunctional components of increasingly complex shapes, reduced material waste and lead time of new products has pushed research into the development of new manufacturing processes [1], e.g., additive manufacturing (AM) technology, which is increasingly applied in the aircraft industry [2,3] due to resulting benefits such as reduced warehousing, inventory management and overall supply chain costs. According to the ASTM Standard F2729-12 a [4], additive manufacturing can be defined as 'the process of joining materials to make objects from 3D model data, usually layer upon layer, as opposed to subtractive manufacturing methodologies, such as traditional machining'. The AM processes were developed in the 1980s as a solution for quicker product development. They were called rapid prototyping, and aimed at producing three dimensional models or mock-ups in order to verify aesthetic and functional performances. In the last twenty years, digital manufacturing of metallic components produced directly from electronic data based on layer-by-layer fabrication has reached a high technological maturity, and is now considered a new production technology named additive manufacturing [5–7]. AM techniques are versatile, flexible, highly customizable and allow the making of parts using a wide variety of materials, and accordingly,

AM can suit most sectors of industrial production [8–10]. Among the AM technologies, the electron beam melting (EBM) and selective laser melting (SLM) manufacturing processes are the most promising. The EBM was designed to process titanium alloys and in particular the Ti6Al4V alloy, as well as materials that require elevated process temperatures [11–14]. The advantages of using EBM in the aerospace engineering and its application especially for the industrialization of aircraft engine were analyzed in [15]. Concerning the Ti6Al4V, the EBM process shows several advantages, such as a fine resultant microstructure, very low residual stress, good static mechanical properties and no oxygen contamination (thanks to the vacuum environment in which the EBM process occurs) [16]. The Ti6Al4V processed by EBM shows a Widmanstattën α + β microstructure [17] much finer than those ones seen in normally processed titanium [18,19]. The reason of a very fine α + β microstructure is due to the rapid cooling of the material melt pools, as in the EBM process the melting and solidification takes places in a matter of seconds. Because of this fine microstructure and the low content of oxygen contamination, static mechanical properties such as the tensile strength of the EBM processed Ti6Al4V are comparable to those of normally processed titanium [20–22]. While the EBM microstructure shows the α-phase grain structure and β-boundary areas, the SLM microstructure consists of α′-martensite platelets. The EBM microstructure shows columnar grain boundaries generally parallel to the build direction. The α′-martensite forms in preference to the acicular α-phase because of the more rapid solidification in the SLM processing in contrast to EBM processing [22–24]. Data of tensile properties are abundant in scientific literature [13,17,22,23] and it is easy to note that Ti6Al4V processed by EBM is characterized by a slight anisotropy showing better elongation at break for the samples built in the horizontal orientation as compared to the vertical ones [25]. On the contrary the properties of tensile strength, yield strength and Young modulus have not shown significant differences. The tensile strength of the SLM specimens were similar to those of the EBM specimens even if the latter were slightly lower than the former ones [13,22]. Investigation on the process parameters aimed at improving Ti6Al4V microstructure produced by EBM were carried out in [26]. The increase of electron beam scanning speed did not show significant effects on the orientation of the grains in the z-plane, on the contrary in the x-y plane significant changes in the preferred orientation were appreciated. The microstructure evolution due to the increase of electron beam scanning speed significantly reduced anisotropy in such properties as hardness and elastic modulus. The evaluation of fatigue properties is critical to understand the behavior of the Ti6Al4V EBM produced parts under cyclic loading. Unlike the large amount of literature on tensile data, published values on the fatigue and fracture performances of Ti6Al4V EBM are limited [18]. The fatigue performance of EBM produced Ti6Al4V samples are lower if compared to the SLM produced Ti6Al4V [22] and to the Metallic Materials Properties Development and Standardization (MMPDS) data [27]. Moreover, the cycle to failure of the fabricated Ti6Al4V samples is significantly lower than the Ti6Al4V wrought, and than the samples obtained by machining Ti6Al4V cylindrical bars produced by EBM [28]. This phenomenon can be explained by the high surface roughness of Ti6Al4V samples produced by EBM, with typical mean roughness ranges between 20 and 30 μm [29]. Contrary to what is noted for the elongation at the break, the anisotropy of the fatigue behavior consists in a significant lower strength for samples built in the vertical orientation [28,30]. Even fracture toughness shows a significant anisotropy due to the different propagation of the crack in samples built in the horizontal and vertical orientation. In [31–33], it is highlighted that the samples built in the vertical orientation and with horizontal notch (i.e., parallel to the building layer) show a significantly worse fracture toughness value ($K_{IC}$) than those fabricated in horizontal orientations. Beginning with the results coming from the analysis of the literature, this study aims at contributing to enlarge the comprehension of the mechanical behavior of Ti6Al4V processed by EBM, providing data of specific mechanical tests and at the same time stimulating a discussion on the possible post process that could improve the material performance.

## 2. Major Issues of EBM Use for Aircraft Application

Considering that Ti6Al4V is one of the more utilized materials in the biomedical and aerospace fields, it is known that the EBM process shows its advantage for all those applications where the need to produce complex shaped parts with conventional techniques leads to a high waste of material. However, the adoption of EBM as production technology in the aerospace sector still meets obstacles due to the complex certification procedure. These include problems concerning the part quality, process repeatability, mechanical properties and above all the low fatigue life. Thereby, it is necessary to demonstrate that parts produced by EBM meet a predetermined set of physical, mechanical and chemical properties [34]. Certifying a batch of conventionally produced material (e.g., wrought or cast material) is a relatively straight-forward and well-established practice in the manufacturing industry since a lot of specific standards, such as the ASTM, ISO or AMS are available. On the other hand, the additive manufacturing industry still lacks a standard for test and quality control methods aimed at certifying additive manufactured parts [35]. For this reason, more studies on the EBM process need to be carried out, with the objective of gathering a set of data useful to create a scientific background and to develop the necessary best practice for designers and production specialists. The objective of this work is to present the results of the mechanical test campaign carried out on Ti6Al4V specimens produced by EBM. Tests were performed on different manufacturing conditions in order to investigate the effect of post processing as machining on mechanical behavior and the effect of the build orientations as well. Results have compared with standard mechanical performance values of Ti6Al4V processed by conventional manufacturing processes. A discussion on test results and on the effect of post processing is provided in order to enlarge the knowledge on viable methods aimed at improving mechanical properties of Ti6Al4V processed by EBM technology.

In this scenario, the next generation of general aviation aircraft (that are subjected in Europe to CS-23 Regulation) is considered as one of the main arenas for the AM technologies. That is mainly due to the potential cost reduction benefit on the manufacturing non-recurrent costs reduction associated to the geometrical complex parts. On the other hand, the potential of the AM technology is facing the "conservative nature" of the aeronautic sector where new technologies have to be evaluated against safety before achieving the any flight clearance. Today, no specific regulation such as the standard CS/FAR regulation paragraphs or specific Acceptable Means of Compliance has been issued by Authority yet. EASA has been working, in conjunction with industry and other regulators, to find the most efficient means by which future regulation of the technology and its applications can be achieved. Following the Certification Memorandum for Additive Manufacturing issued by EASA at the end of 2017 [36], it is essential that design values used for AM materials reflect not only the variability of the constituent materials as purchased by the suppliers, but also the variability introduced by the manufacturing process used to fabricate production parts. AM variability is to be shown to be controlled through material specifications in combination with process controls defined in process specifications, including post processing operations. These specifications (for both, material and process) as well as the method(s) of manufacture, shall be introduced in the type design under the design approval applicant or holder responsibility. The current practice establishes that any aeronautical flyable AM application is treated case by case and requires to advise and to involve the authority at the earliest stage during the development and implementation of AM. For that, the applications in the general aviation sector are only limited and the main advancements are still part of specific research projects. These projects are aimed to promote and grow AM technology in the aviation sector providing a set of data on mechanical and microstructural properties required by authorities to aid the comprehension of the AM process (dependent on specific manufacturer practice), its limits and its level of reliability. One of the open scientific streams to be covered is the experimental material/process characterization, data variability and the effects of post-processing action on the final performances. All of these points still remain open, and are deemed crucial for a possible exploitation after authorities' approval. Even if many other works have been carried out on the mechanical characterization of Ti6Al4V produced by EBM with and without post-processing, the current study differs from the others, due to its specific

objective. It wants to give additional knowledge to general aviation (GA) aircraft development current practice, concerning use of a new (with respect GA specific sector) manufacturing process. In more detail, the authors have identified the basic data to support the GA structural substation and the design loop (i.e., mechanical characteristics and their statistics variability). In addition, clear indication on how the post-processing reworks can improve the structural performances of any primary structure element (PSE) has been given by the authors. Although, the rework can affect the total recurrent cost and the delivery time of the PSE, contributing to decrease the AM specific technology appeal. It is deemed crucial to have quantified the benefit of rework in terms of the mechanical proprieties' improvement and to supply designers the suitable quantitative data to drive their decisions. All those considerations can be considered as a very first application for a GA PSE.

## 3. Materials

Raw materials used in EBM are metallic particles obtained from powder metallurgy. Some powder features strongly affect the process performance. The main factors in EBM affecting processing conditions are the flowability, powder packing and the heat transfer process phenomena. The powder used in EBM is spherical in shape because contributes to improve flowability and, thus, may ensure high build rates and part accuracy. In general, fine powder is used in EBM. The powder size distribution also has a significant effect on the build part density, surface finish and mechanical properties [37]. The samples used in this study were manufactured by using Ti6Al4V atomized powder with spherical morphology. The spherical shape may contribute to improve flowability and may ensure high build rates and accuracy [38]. The powder flow rate measured according to [39] was found as 25 s/50 g. The apparent density according to [40] was 2.57 g/cm$^3$. As regarding the particle size distribution, the percentage by mass of particle size in the range 45–106 μm was found equal to 93.7%, the percentage by mass of particle size, measured according to [41], is shown in Figure 1. The powder nominal chemical composition is summarized in Table 1. The whole Ti6Al4V powder characterization was provided by the supplier (ARCAM Company, Gothenburgh, Sweden) with a certificate of analysis. Such a document contains for the supplied batch of raw material, the results of the powder characterization in according to the above mentioned ASTM tests; i.e., the values of the powder flow rate, the apparent density, the particle distribution size and the chemical composition.

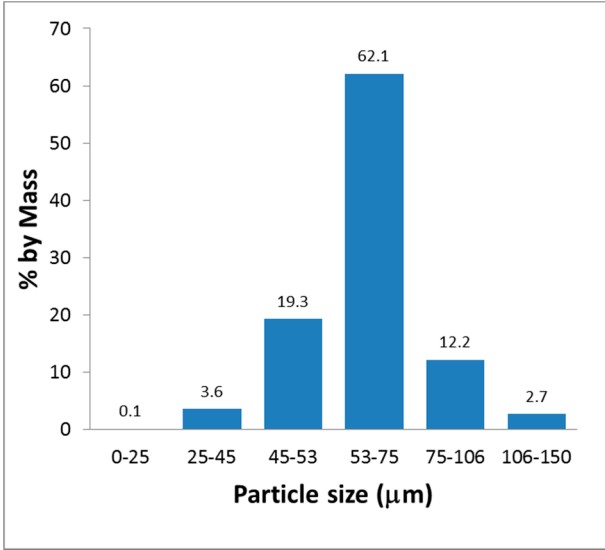

**Figure 1.** Particle size distribution according to ASTM B214.

**Table 1.** Nominal chemical composition in percentage of weight of the Ti6Al4V pre-alloyed powder used in electron beam melting (EBM).

| Chemical Element | % | Required % ASTM F2924 |
|---|---|---|
| Al | 6.40 | 5.50–6.75 |
| V | 4.12 | 3.50–4.50 |
| Fe | 0.18 | <0.30 |
| O | 0.14 | <0.20 |
| N | 0.01 | <0.05 |
| H | 0.003 | <0.015 |
| C | 0.01 | <0.08 |
| Ti | Balance | Balance |

## 4. Methods

In this study an ARCAM A2X EBM machine was used and the software installed was the EBM Control version 3.2 (ARCAM, Gothenburgh, Sweden). The specimens manufactured were built by using Ti6Al4V build themes consisting in the 50 µm ARCAM standard process parameters with building layer thickness of 50 µm. The tests performed to mechanically characterize the Ti6Al4V produced by EBM were carried out on two types of manufacturing conditions:

- "As built" directly EBM manufactured in the dog-bone shape suitable to be tested
- "Machined" obtained by machining cylindrical bars produced by EBM

A test matrix showing samples information as type manufacturing condition, build orientation and the related standard used is summarized in Table 2.

**Table 2.** Test matrix and the related standard used.

| Type of Test | Sample Type | Standard | Manufacturing Condition | Build Orientation | N. of Sample | Remark |
|---|---|---|---|---|---|---|
| Tensile Test | Cylindrical Standard dimensions | ASTM E08 | As built | 0°; 45°; 90° | 9 | n. 3 sample for each build orientation |
| Tensile Test | Cylindrical Standard dimensions | ASTM E08 | Machined | 0°; 45°; 90° | 9 | n. 3 sample for each build orientation |
| Fatigue Test | Cylindrical Standard dimensions | ASTM E466 | As built | 90° | 30 | n. 5 sample for n. 6 point of the Wohler curve |
| Fatigue Test | Cylindrical Standard dimensions | ASTM E466 | Machined | 90° | 30 | n. 5 sample for n. 6 point of the Wohler curve |
| Fracture Toughness | Compact Tension | ASTM E399 | Machined | 90° and horizontal notch | 6 | n. 6 compact CT samples for 90° build orientation only |

The compact tension specimens were built in vertical orientation and with a horizontal notch (i.e., parallel to the building layer). Moreover, the samples identified in Table 2 with the term "machined" were post-processed by machining. Such a post processing was carried out by CIRA's industrial partners. The machining parameters were compliant to those ones commonly used for cast Ti6Al4V.

### 4.1. Tensile Tests

Round tensile samples were produced by EBM technology, with final geometry and dimensions compliant to the standard ASTM E08 [42] as shown in Figure 2.

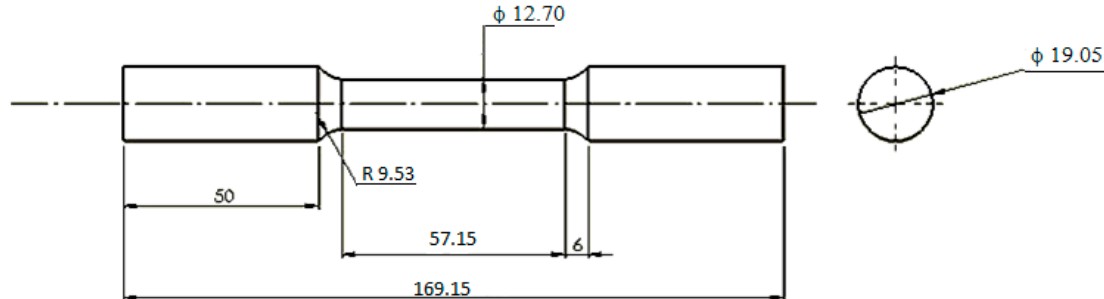

**Figure 2.** Drawing of tensile specimens, the expressed dimensions are referred to mm.

The specimens were manufactured in two manufacturing conditions ("as built" and "machined" as even shown in Figure 3.

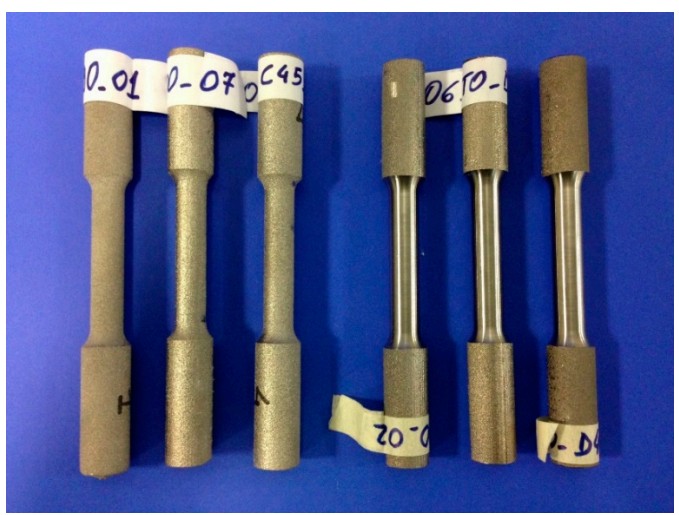

**Figure 3.** Three as built (on the left side) and three machined specimens after EBM process (on the right side).

For each manufacturing conditions (i.e., "as built" and "machined"), three sets of specimens were produced:

- n.3 tensile samples with 0° build orientation with respect to the start plate (x-y plane);
- n.3 tensile samples with 45° build orientation with respect to the start plate (x-y plane);
- n.3 tensile samples with 90° build orientation with respect to the start plate (x-y plane).

In the Figure 4a drawing that explains how the above mentioned three sets of specimens were built and located inside the build envelope of the EBM machine is shown.

The tensile tests were performed by using an MTS250 machine with the load cell of 250 kN. The percentage strain-to-failure was measured using a clip-on extensometer that was attached to the gage section of the test specimens.

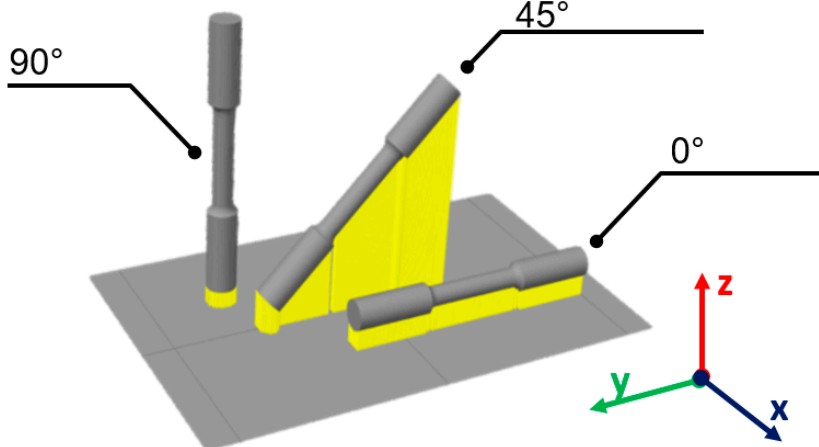

**Figure 4.** Drawing explaining the build orientation of the specimens at 0°,45°,90° respect the start plate (x-y plane).

*4.2. Fatigue Tests*

Two sets of Ti6Al4V specimens were manufactured and tested to fatigue. Both the sets of specimens were produced with 90° build orientation with respect to the start plate (x-y plane), that is in the vertical direction. In particular were EBM-manufactured:

- N. 25 machined specimens, i.e., produced by machining cylindrical bars manufactured by EBM process;
- N. 25 specimens as built, i.e., produced by EBM in the dog-bone shape suitable to be tested.

The configuration with build orientation at 90° with respect to the start plate was chosen because, in according to [28], it is to be considered the weakest fatigue build orientation, i.e., that one related to the minimum fatigue properties.

Geometry and dimensions of all the specimens are compliant to the standard ASTM E466 [43] as shown in the Figure 5, and the images of the specimens ("as built" and "machined") are shown in the Figure 6.

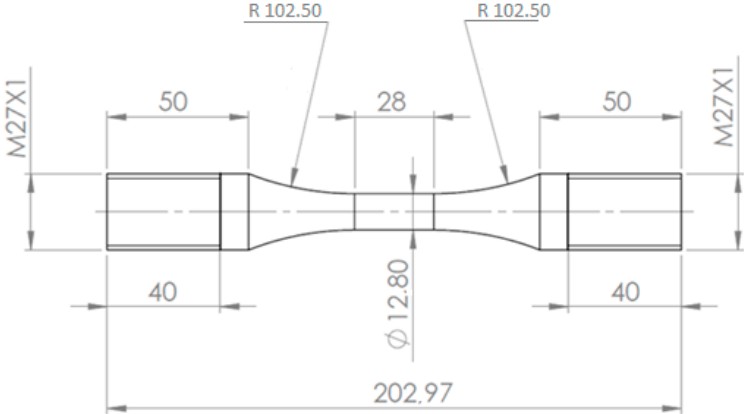

**Figure 5.** Drawing of fatigue specimens, the expressed dimensions are referred to mm.

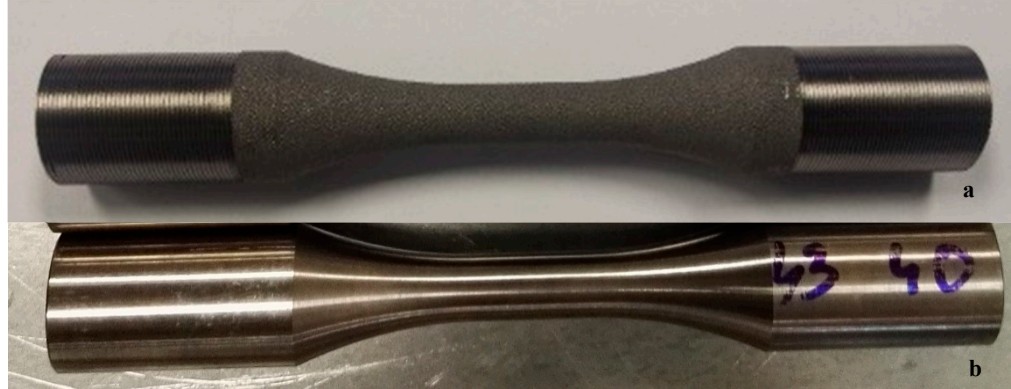

**Figure 6.** (**a**) Photo of as built specimen; (**b**) photo of machined specimen.

The fatigue tests were performed by using a Zwick/Roel AMSLER100 machine with a load cell of 100 kN and the test execution parameters R = 0.1 and the load frequency = 87 Hz.

### 4.3. Linear-Elastic Plane-Strain Fracture Toughness $K_{IC}$ Tests

Linear-elastic plane-strain fracture toughness $K_{IC}$ tests, compliant to the ASTM E399 [44], were performed on n.6 compact tension (CT) samples whose geometry and its build orientation are shown in Figure 7.

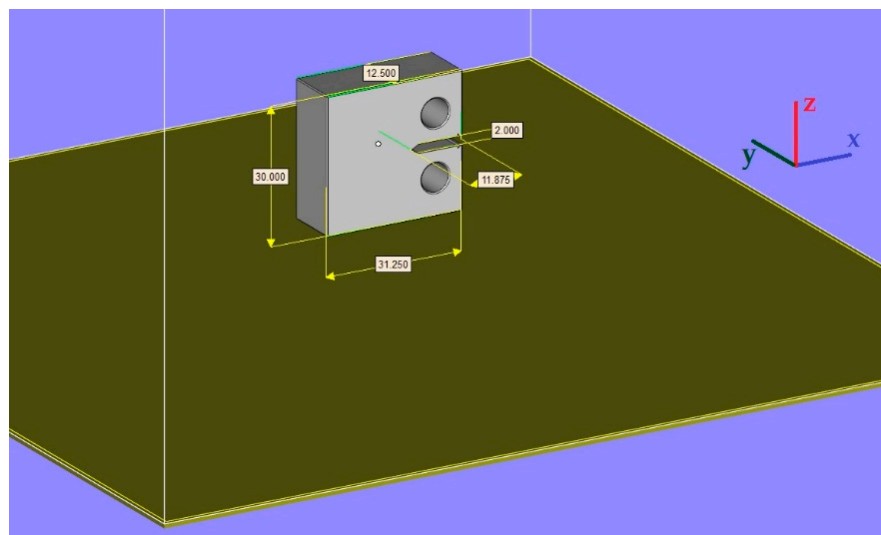

**Figure 7.** Drawing of compact tension (CT) sample, the dimensions are referred to mm.

The samples were obtained by machining the external surfaces of n.6 parallelepipeds (35 × 35 × 15 mm) produced by EBM technology (Figure 8a). The biggest side of the parallelepiped was oriented parallel to the z axis and the notch was made along the horizontal direction (i.e., parallel to the building layer as shown in the sketch of Figure 7. This configuration was chosen because, according to [31–33], it is to be considered the weakest fracture toughness build orientation, i.e., that one related to the minimum $K_{IC}$ value. The notch was made by wire EDM (Electrical Discharge Machining). The final shape of CT specimens obtained by machining the 35 × 35 × 15 mm parallelepipeds produced by EBM is shown in Figure 8b.

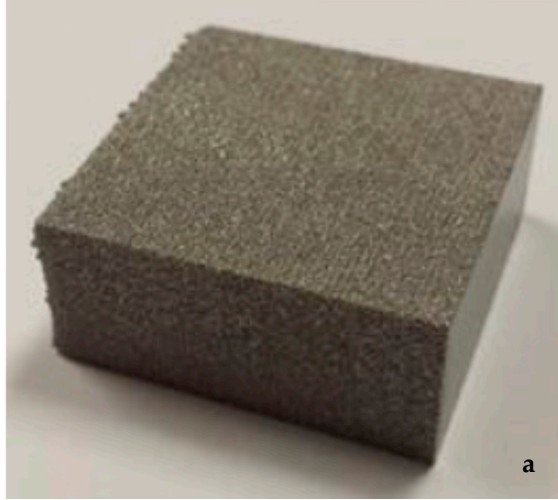 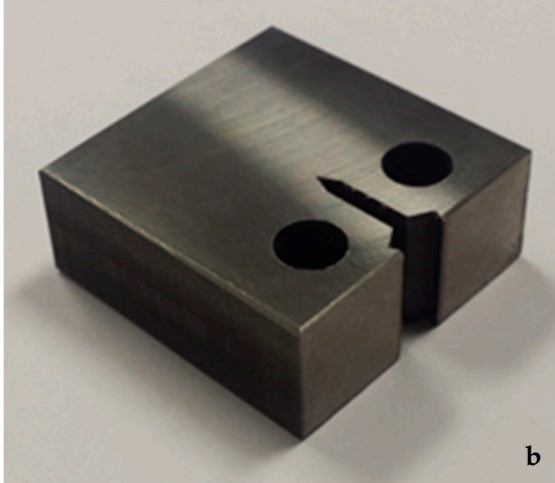

**Figure 8.** (**a**) parallelepiped $35 \times 35 \times 15$mm produced by EBM; (**b**) CT samples obtained by machining parallelepiped of (**a**).

The linear-elastic plane-strain fracture toughness $K_{IC}$ tests were performed by using an Instron 8801 machine with load cell of 100 kN. The displacement gage was measured by a double cantilever clip on extensometer that was attached to the specimens by knife-edges.

## 5. Results

In the following sub-paragraph the results of the performed mechanical tests, as described in the test matrix (Table 2), are reported. The obtained performances of Ti6Al4V processed by EBM technology were compared with those ones of Ti6Al4V produced by conventional manufacturing processes.

### 5.1. Tensile Tests

The tensile tests were performed in compliance to the ASTM E08. The test results are summarized in Figure 9, in which a graphic representation of the plot of average and standard deviation of the tensile properties is shown.

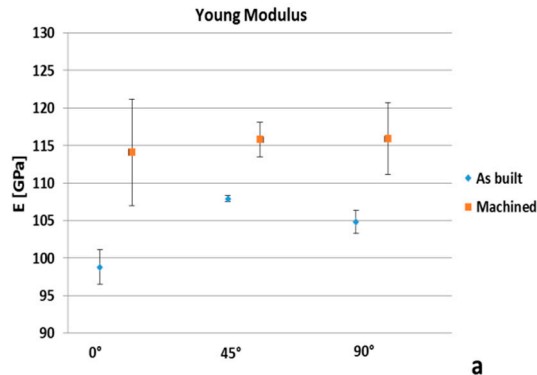

**Figure 9.** *Cont.*

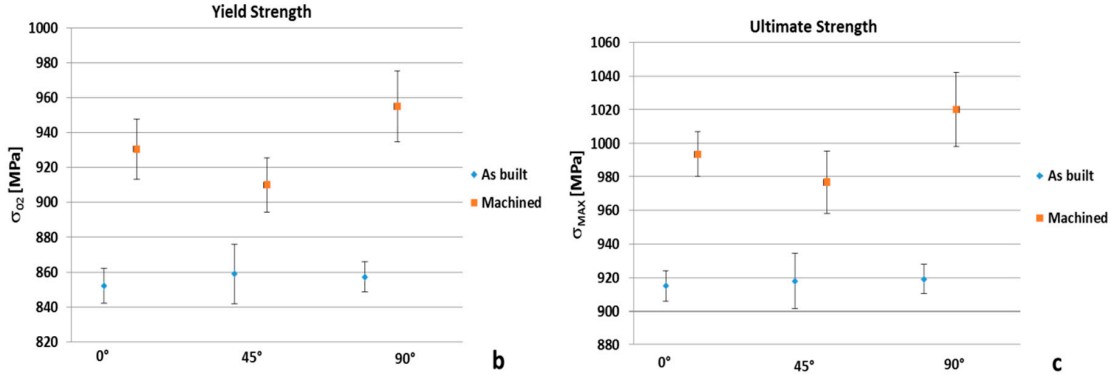

**Figure 9.** Plots of average and standard deviations: (**a**) Young modulus values; (**b**) yield strength values; (**c**) ultimate strength values.

In Table 3 it is shown the tensile data of the "as built" set of specimens. In this table it is shown the tensile properties values of each tested specimen (i.e., ultimate strength σmax, yield strength $\sigma_{02}$ and young modulus E) and even a statistic of all the tensile properties (i.e., average, standard deviation, and relative standard deviation) gathered for each build orientation.

**Table 3.** Test data of as built specimens and the related statistics.

| ID of the Specimen | Build Direction | E [GPa] | $\sigma_{02}$ [MPa] | Σmax [MPa] |
|---|---|---|---|---|
| spec 1 as built | | 101.4 | 864.2 | 925.7 |
| spec 2 as built | 0° | 97.5 | 845.7 | 910.5 |
| spec 3 as built | | 97.4 | 848.0 | 908.8 |
| Average | | 98.8 | 852.6 | 915.0 |
| ST.DEV | | 2.3 | 10.0 | 9.3 |
| RELATIVE ST.DEV | | 2.3% | 1.2% | 1.0% |
| spec 4 as built | | 108.0 | 872.8 | 932.5 |
| spec 5 as built | 45° | 107.5 | 864.6 | 921.5 |
| spec 6 as built | | 108.2 | 840. 0 | 900.0 |
| Average | | 107.9 | 859.1 | 918.0 |
| ST.DEV | | 0.4 | 17.1 | 16.5 |
| RELATIVE ST.DEV | | 0.3% | 2.0% | 1.8% |
| spec 7 as built | | 106.0 | 865.7 | 927.8 |
| spec 8 as built | 90° | 105.4 | 857.6 | 918.7 |
| spec 9 as built | | 103.1 | 848.6 | 910.8 |
| Average | | 104.8 | 857.3 | 919.1 |
| ST.DEV | | 1.5 | 8.6 | 8.5 |
| RELATIVE ST.DEV | | 1.5% | 1.0% | 0.9% |

In Table 4 it is shown the tensile data of the "machined" set of specimens. In this table it is shown the tensile properties values of each tested specimen (i.e., tensile strength σmax, yield strength $\sigma_{02}$ and young modulus E) and even a statistic of all the tensile properties (i.e., average, standard deviation, and relative standard deviation) gathered for each build orientation.

**Table 4.** Test data of machined specimens and the related statistics.

| ID of the Specimen | Build Direction | E [GPa] | $\sigma_{02}$ [MPa] | $\Sigma$max [MPa] |
|---|---|---|---|---|
| spec 1 machined | | 122.3 | 935.4 | 981.9 |
| spec 2 machined | 0° | 110.7 | 944.7 | 990.3 |
| spec 3 machined | | 109.4 | 911.3 | 1008.0 |
| Average | | 114.1 | 930.5 | 993.4 |
| ST.DEV | | 7.1 | 17.2 | 13.3 |
| RELATIVE ST.DEV | | 6.2% | 1.9% | 1.3% |
| spec 4 machined | | 113.2 | 897.3 | 955.3 |
| spec 5 machined | 45° | 117.3 | 927.2 | 986.7 |
| spec 6 machined | | 116.9 | 905.3 | 988.3 |
| Average | | 115.8 | 909.9 | 976.8 |
| ST.DEV | | 2.3 | 15.5 | 18.6 |
| RELATIVE ST.DEV | | 2.0% | 1.7% | 1.9% |
| spec 7 machined | | 120.0 | 978.1 | 1041.6 |
| spec 8 machined | 90° | 110.6 | 938.0 | 997.6 |
| spec 9 machined | | 117.0 | 949.0 | 1021.0 |
| Average | | 115.9 | 955.0 | 1020.1 |
| ST.DEV | | 4.8 | 20.7 | 22.0 |
| RELATIVE ST.DEV | | 4.1% | 2.2% | 2.2% |

*5.2. Fatigue Tests*

The stress-cycles curves (σ-N cσurves), according to the ASTM E466, were calculated for each set of specimens. For the sake of brevity, in Figure 10 the σ-N curves for all the type of specimens tested are reported on the same graph, in addition it is reported even the σ-N curve of standard Ti6Al4V in annealed condition [45].

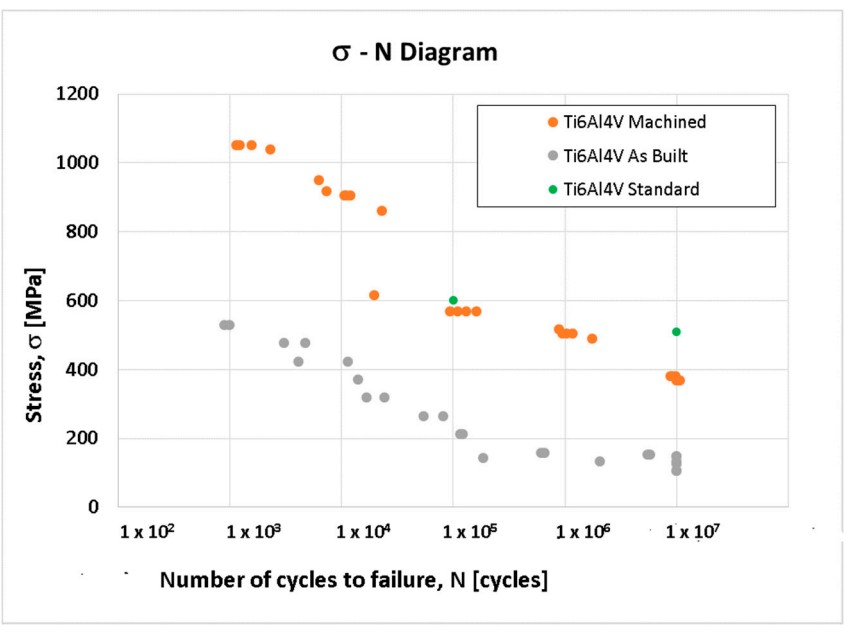

**Figure 10.** Plot of σ-N Curves.

The results shown in the Figure 10 demonstrate that as expected, the fatigue behavior of as built samples is significantly worse than those ones of machined samples.

Investigations are carried out on the surface fractures of both the type of specimens ("as built" and "machined") showing that in both the cases cracks originates at surface (Figures 11 and 12).

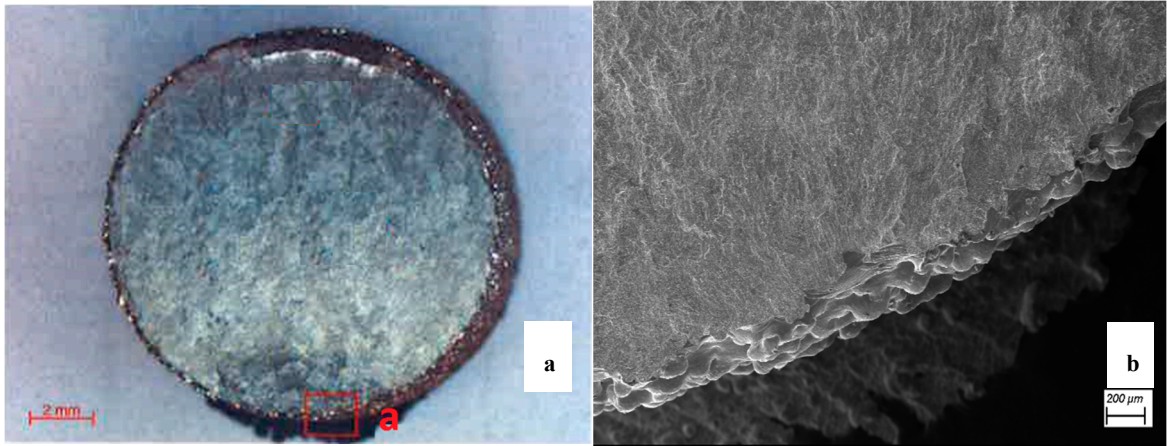

**Figure 11.** Macrograph of fracture surface of fatigue tested "as built" sample: (**a**) image of the surface at 7× magnification; in the "a" area is located the crack nucleation; (**b**) image of the crack nucleation area at 100× magnification.

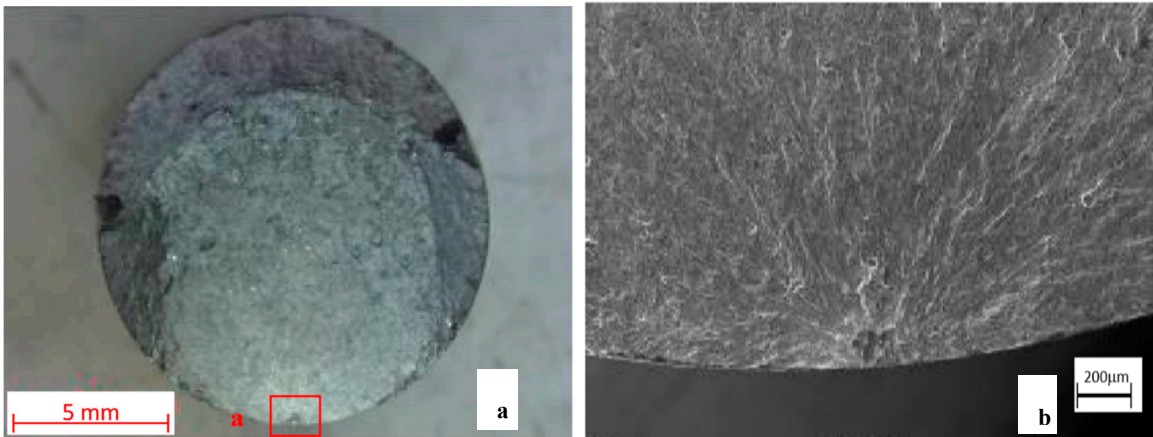

**Figure 12.** Macrograph of surface fracture of fatigue tested "machined" sample: (**a**) image of the surface at 7× magnification; in the "a" area is located the crack nucleation; (**b**) image of the crack nucleation area at 100× magnification.

*5.3. Linear-Elastic Plane-Strain Fracture Toughness KIC Tests*

Linear-elastic plane-strain fracture toughness $K_{IC}$ tests were carried out in according with ASTM E399. The results in terms of $K_{IC}$ values are shown in the Table 5.

**Table 5.** Results of fracture toughness test.

|  | CT1 | CT2 | CT3 | CT4 | CT5 | CT6 | MEAN | St. Dev. |
|---|---|---|---|---|---|---|---|---|
| $K_{IC}$ [MPa $\sqrt{m}$] | 29.6 | 33.4 | 31.2 | 30.2 | 32.6 | 30.5 | 31.25 | 1.47 |

A typical graph force–crack mouth displacement is reported in the Figure 13.

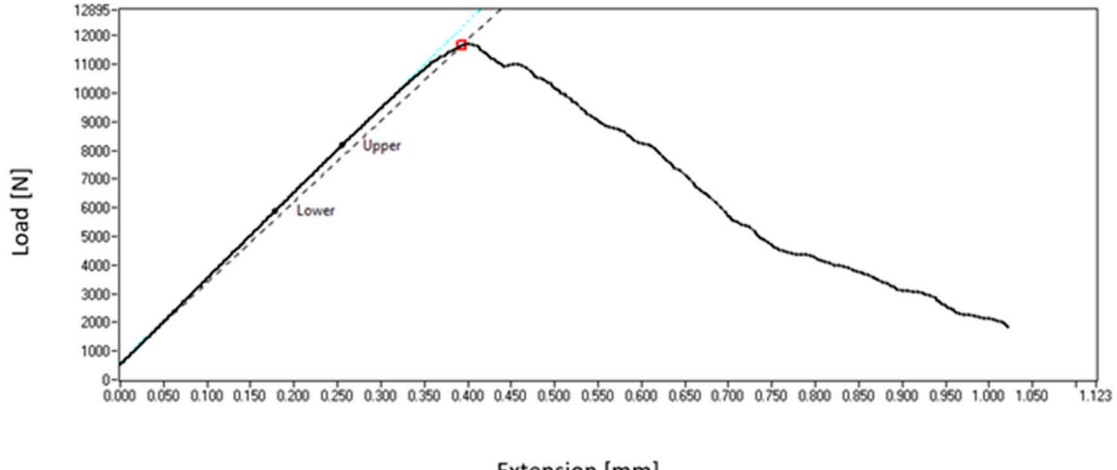

**Figure 13.** Typical curve force–crack mouth displacement of a CT sample.

An optical analysis of the fracture surfaces clearly highlights that no plastic deformation is observed showing an evident brittle fracture mode (Figure 14). For this reason the linear-elastic plane-strain fracture toughness tests are guaranteed to be resulted in a $K_{IC}$ value measured under plane-strain conditions.

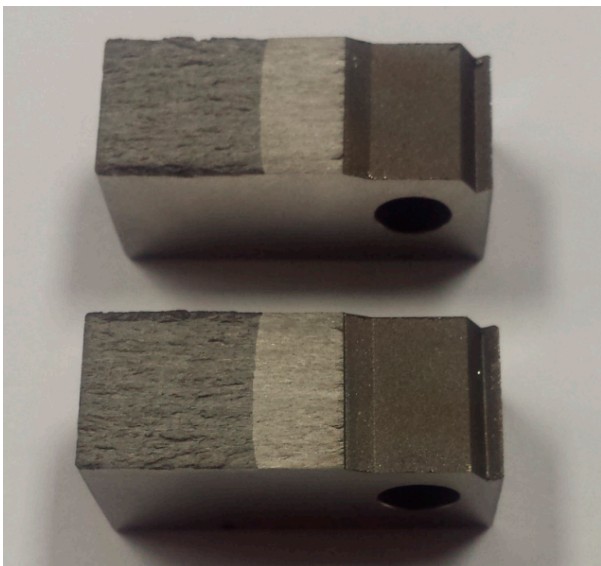

**Figure 14.** Typical fracture surface of a CT sample.

Ti6Al4V processed by EBM shows a $K_{IC}$ value less than the half of $K_{IC}$ value of Ti6Al4V standard (in annealed condition) [45] which is 74.6 [MPa·m$^{1/2}$].

## 6. Discussion

This section has a discussion on the results of the tensile, fatigue and linear-elastic plane-strain fracture toughness $K_{IC}$ tests.

The tensile test results have shown a very high repeatability, as even confirmed by their small values of the standard deviations and of the relative standard deviations calculated in Tables 3 and 4. For this reason it is possible to affirm that the tensile data resulting from this work can be considered statistically reliable.

Comparing the test results with the tensile performance of annealed Ti6Al4V (i.e., tensile strength 950 MPa; yield strength 880 MPa; Elongation at break 25%; Young Modulus 113.8 GPa [45]), it is possible to assert that:

- Tensile properties revealed that the specimens tested from the 90° built samples have marginally higher strength values if compared to the specimens built in the 0° and 45° orientation. Nevertheless, the differences fall within the normal statistical dispersion of the data so that the tests do not show a significant anisotropy of the material as even shown by the plots represented in Figure 9;
- Ultimate strength, yield strength, and Young modulus of Ti6Al4V specimens obtained by machining bar produced by EBM show slight better performance than standard Ti6A4V (annealed condition);
- Ultimate strength, yield strength, and Young modulus of Ti6Al4V specimens in "as built conditions" show a slight worse performance than standard Ti6A4V (annealed condition);
- Machined specimens show a high Young modulus homogeneity among build orientations and significant higher tensile performances values if compared with as built specimens.

The lower values of tensile properties (ultimate strength, yield strength and Young modulus) of as built specimens compared to the machined ones are mainly due to significant reduction of the actual cross section caused by the high irregular external circular surfaces, as confirmed by the literature data on this topic [46,47].

The test campaign was carried out to characterize the fatigue and fracture toughness properties of Ti6Al4V produced by EBM in both manufacturing conditions (as built and machined) was performed, for the sake of brevity, only on the configuration of the 90° build orientation in respect to the start plate. This orientation was chosen because it represents the weakest build orientation, i.e., that one related to the minimum fatigue and fracture toughness properties.

It is well known from literature that bad fatigue performance is mainly due to the high surface roughness and/or high internal porosity [48]. The fatigue cracks, as shown from the surface fracture analysis in the Figures 11 and 12 for both types of specimens (as built and machined) initiate at the surface in the homogeneous materials. The high roughness of as built specimens surfaces lead to a quick nucleation of the fatigue crack causing a significantly reduction of the fatigue life. The fatigue curves (σ-N) demonstrate that the fatigue strength of the "Ti6Al4V as built", because of the very high surface roughness, is significantly worse than the "Ti6Al4V machined one". Moreover, the machining generates on the specimen surface residual stresses that can retard the crack nucleation, improving the fatigue behavior of the machined Ti6A4V specimens [49]. The σ-N curve of the "Ti6Al4V machined" shows a fatigue behavior comparable with that of the Ti6Al4V standard (in annealed condition) [45], even if the former does not reach a fatigue limit and seems to be slightly lower than the latter one.

As reported in Table 5, Ti6Al4V processed by EBM shows a $K_{IC}$ value of 31.25 [MPa·m$^{1/2}$] with a small standard deviation (1.47 [MPa·m$^{1/2}$]) which confirms the reliability of the test results. The $K_{IC}$ calculated for Ti6Al4V processed by EBM is less than the half of $K_{IC}$ value of Ti6Al4V standard (in annealed condition) [45] which is 74.6 [MPa·m$^{1/2}$]. The analysis conducted on the fracture surfaces shows that no plastic deformation is noted (Figure 14) and that the $K_{IC}$ value calculated by the tests is measured under plane-strain conditions. A possible explanation of such a low value of $K_{IC}$ can be found in the high level of defects consisting in voids, porosity, and unmelted particles due to lack of fusions [32] which can considerably reduce the fracture toughness. In their study Seifi et al. [31], metallographic cross sections analysis of the fracture surface of CT tested specimens produced by EBM revealed the presence of defects. In more detail, the larger defects were always perpendicular to the build direction (i.e., parallel to the building layer).

The Ti6Al4V components produced by EBM present several issues which need to be still investigated: static tensile anisotropy, high irregular external circular surface which significantly reduces the tensile properties, high roughness which reduces the fatigue behavior and internal defects

which reduce the fatigue strength and fracture toughness. Many studies have been conducted on machining and on hot isostatic process as suitable post-processing in order to reduce respective surface roughness and internal defects such as voids and porosity [50,51].

All these aspects should be taken into account when EBM is used to manufacture aerospace components. Further research should follow, on the possible post-processing in order to improve the mechanical properties of Ti6Al4V components produced by EBM. The materials to be employed in aerospace applications have to meet strict requirements in terms of mechanical performances-to-weight ratios and in terms of reliability.

The aim of this work was to investigate the main issues concerning the EBM process and its influence on Ti6Al4V mechanical properties. The results of this study have pointed out that the mechanical performances of Ti6Al4V produced by EBM are influenced by many factors. The discussion on the mechanical characterization and the possible post-processing improving the mechanical performances of Ti6Al4V processed by EBM is a contribution to increase the comprehension of the EBM process. At the same time it provides useful information to designers in order to improve their confidence in components designing, and to technologists in order to choose better EBM manufacturing strategies.

## 7. Conclusions

The results shown in this work were developed in the framework of Clean Sky 2 Small Aircraft more affordable Manufacturing project (SAT-AM), based on a consortium with a relevant track records for this size of aircraft [52–55], within which will be developed a primary structure element for general aviation application made in AM.

In particular, a mechanical characterization of Ti6Al4V processed by EBM technology was performed in order to compare its performance with Ti6Al4V standard (in annealed condition). Ti6Al4V EBM processed was tested in two different manufacturing conditions "as built" and "machined" (i.e., post-processed by machining).

The results can be summarized:

-   The tensile tests have shown high mechanical performance of Ti6Al4V EBM-processed, particularly the specimens obtained by machining bars produced by EBM have shown tensile results slightly better than standard Ti6Al4V in annealed condition [45];
-   The fatigue performance of Ti6Al4V processed by EBM are generally lower than Ti6Al4V standard (in annealed condition) [45], highlighting, on the other side, that Ti6Al4V produced by EBM in as built condition shows the worst fatigue behavior;
-   The stress-cycles curve of Ti6Al4V obtained by machining cylindrical bars produced by EBM shows a comparable, even if slightly lower, behavior if compared with Ti6Al4V standard (in annealed condition) [45];
-   The $K_{IC}$ value of Ti6Al4V produced by EBM is considerably worse than Ti6Al4V standard (in annealed condition) [45].

This study leads to focus on aspects concerned with the EBM process issues which significantly affect the mechanical properties of materials produced by this manufacturing technology. At the same time it provides information and considerations useful to direct further investigations on the possible post-processing strategies aimed at improving mechanical performance of Ti6Al4V processed by EBM technology. Mechanical characterization and investigations on process and post-process strategies are needed, to make it possible that Ti6Al4V processed by EBM should reach the mechanical performances required for many components employed in aerospace applications. The performed characterization of EBM processed Ti6Al4V is preparatory for the design and flight clearance achievement of the PZL M28 modified aircraft. In particular the test campaign and the comforting results in terms of low mechanical performance variability, has convinced the researchers to develop a primary specific structural element (i.e., nose landing gear attachment) for a CS-23 aircraft to be cleared for flight. This study is considered

quite innovative for this aircraft category, since no many other scientific works have discussed the issue of how to approach research activities preparatory for AM certification in the aerospace sector.

**Author Contributions:** Conceptualization, C.P., S.F. and R.B.; formal analysis, C.P., S.F. and R.B.; methodology, C.P., S.F. and R.B.; software, C.P., S.F. and R.B.; investigation, C.P., S.F. and R.B.; data curation, C.P., S.F. and R.B.; writing—reviewing, C.P., S.F. and R.B., A.C. and L.D.P.; supervision, A.C. and L.D.P., project administration, L.D.P. and A.C.; funding acquisition, L.D.P. All authors have read and agreed to the published version of the manuscript.

**Funding:** This research was funded by H2020 Clean Sky 2 Framework (CS2), 807083—AIRFRAME ITD (GAM AIR 2018), Topic Identification Code: JTI-CS2-2015-CPW02-AIR-02-07, project number and acronym: 699757/SAT-AM (More Affordable Small Aircraft Manufacturing) and the APC was funded by H2020 Clean Sky 2 Framework.

**Acknowledgments:** Alessio Faraguti (TEC Euolab srl) is kindly acknowledged for his technical support.

**Conflicts of Interest:** The authors declare no conflict of interest.

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
