# Peer review of "The Effect of Post-Processing on the Mechanical Behavior of Ti6Al4V Manufactured by Electron Beam Powder Bed Fusion for General Aviation Primary Structural Applications"

_aerospace, doi:10.3390/aerospace7060075_

Round 1

Reviewer 1 Report

In general, the paper has numerous grammar mistakes, and many of the sentences are hard to understand as many things want to be mentioned in them. The paper presents the experimental results of AM-metallic samples and compared them to AM-metallic-machined samples. The paper poorly presents the methods, machines, and parameters employed. The paper lacks discussion and further analysis on why the results obtained were obtained. Specific comments/questions to be addressed are included in the list below.

L11 – full-density to fully-dense. Still from AM there is no way one can create a fully-dense component.

Abstract, it takes 8 lines to know what the paper is going to be about, I recommend this to be mentioned in 2nd or max 3rd line of the abstract.

L19 & 78 – “material conditions” the authors mean manufacturing conditions?

L22 – developing -> developed

L47 – remove “really”

L57 to L60 – sentence is too long, making it difficult to understand. Please split it into 2 or 3.

L65 – how similar?

L72 – on the other hand (remove S at the end)

Section 1 – a thorough review of the relevant literature is expected. The are numerous works on AM-metallic samples.

Section 2 – which machine do the authors used? This is a crucial aspect and it was not mentioned in the section.

Section 2 – change Growth orientation to “build orientation”

Section 2 – the reviewer is not sure what do the authors mean with “as built” in column 4 Table 2, if it refers to additively manufactured without post manufacturing process? Additionally, “machined” samples means as a postprocess or does it refers to samples that were manufactured with no AM. Machining parameters and information are missing.

Section 2 – in general is confusing and it lacks a clear description of what was done. Which were the machines used for all the testing presented in Table 2? Can some figures with the samples geometries or experimental setup be included? Some of these data are scattered presented in Section 3. These should be in Materials and methods.

Section 3 – the scale in fig. 2 make the differences in the comparison seemed to significant, however looking closely at the values maybe no so, can the authors present error % and discuss the differences?

Section 3 – The so-called standard points presented in Fig. 3, where do they come from? From the Fatigue results in the SN curve can one conclude an Endurance Limit?

Section 3 -  is called Results and Discussion, however the authors only present the results in a poor way, lacking discussion as to why the differences between AM and AM-machined components have different mechanical properties. How is the building orientation affecting the results?

Author Response

Reviewers 1

In general, the paper has numerous grammar mistakes, and many of the sentences are hard to understand as many things want to be mentioned in them. The paper presents the experimental results of AM-metallic samples and compared them to AM-metallic-machined samples. The paper poorly presents the methods, machines, and parameters employed. The paper lacks discussion and further analysis on why the results obtained were obtained. Specific comments/questions to be addressed are included in the list below.

Grammar was reviewed and even English style was improved. The paper was even completely reviewed in its structure. A more detailed analysis of the literature was done. The section Materials and Methods was expanded with three subsections one for each type of test performed. In these sub-sections it was explained the methods, the machines, the setups used for the testing and information on post-processing (machining). Moreover in these subsections were added several images of samples drawing and photos of the samples pre and after post processing. The section 3 was changed too. Three sub-sections (one for each mechanical test performed) were added. And section 4 was added in order to deeply afford the discussion of tests results from a scientific and speculative point of view.

L11 – full-density to fully-dense. Still from AM there is no way one can create a fully-dense component. done

Abstract, it takes 8 lines to know what the paper is going to be about, I recommend this to be mentioned in 2nd or max 3rd line of the abstract.

done

L19 & 78 – “material conditions” the authors mean manufacturing conditions?

done

L22 – developing -> developed

This sentence was deleted

L47 – remove “really”

done

L57 to L60 – sentence is too long, making it difficult to understand. Please split it into 2 or 3.

It was split and reformulated

L65 – how similar?

This sentence is completely changed in: “the static mechanical properties like for example tensile strength of the EBM processed Ti6Al4V are comparable to those ones of normally processed titanium”

L72 – on the other hand (remove S at the end)

done

Section 1 – a thorough review of the relevant literature is expected. There are numerous works on AM-metallic samples.

Section 1 was completely reviewed. All the information concerning the project was eliminated. A more detailed analysis of the literature was done on Ti6Al4V “produced by EBM”: microstructure and mechanical characterization. In particular for tensile, fatigue and fracture toughness properties of Ti6Al4V processed by EBM was analyzed the effect of the anisotropies and of the defects (porosity, lack of fusions, surface morphology and roughness). The analysis of the state of the art on Ti6Al4V produced by EBM gives the chance of opening a reflection on investigations of possible post-processings in order to improve mechanical performance of the material. At this aim this work try to give information on tensile, fatigue and fracture toughness behavior of Ti6Al4V produced by EBM. The post processings, as well the machining which was investigated in this work, can lead to an improvement of the mechanical properties of the Ti6Al4V produced by EBM. It is deemed crucial for the ALM airworthiness viability.

Section 2 – which machine do the authors used? This is a crucial aspect and it was not mentioned in the section.

Section 2 was completely changed. This section was expanded with three subsections, one for each type of test performed. In these sub-sections it was explained the methods, the machines, the setups used for the testings and information on post-processing (i.e. machining). Moreover in these subsections were added several images of samples drawings and photos of the samples pre and after post processing.

Section 2 – change Growth orientati8uon to “build orientation”

done

Section 2 – the reviewer is not sure what do the authors mean with “as built” in column 4 Table 2, if it refers to additively manufactured without post manufacturing process? Additionally, “machined” samples means as a postprocess or does it refers to samples that were manufactured with no AM. Machining parameters and information are missing.

In section 2  was added (lines 118,119) the following:

“As built” directly EBM manufactured in the dogbone shape suitable to be tested

“Machined” obtained by machining by CNC cylindrical bars originally manufactured by EBM

It was added in this section and the related sub-sections that machining activities were conducted by industrials CIRA’s partners. Moreover the parameters of machining used were compliant to those one commonly used for cast Ti6Al4V.

Section 2 – in general is confusing and it lacks a clear description of what was done. Which were the machines used for all the testing presented in Table 2? Can some figures with the samples geometries or experimental setup be included? Some of these data are scattered presented in Section 3. These should be in Materials and methods.

Section 2 was completely changed. This section was expanded with three subsections, one for each type of  performed test. In these sub-sections it was explained the methods, the machines, the setups used for the testings and information on post-processing (machining). Moreover in these subsections were added several images of samples drawing and photos of the samples pre and after post processing.

Section 3 – the scale in fig. 2 make the differences in the comparison seemed to significant, however looking closely at the values maybe no so, can the authors present error % and discuss the differences?

The section 3 was changed. Three sub-sections (one for each mechanical test performed) were added. In sub section 3.1 two tables (tab.3 and tab.4) were added in which are shown the tensile properties values of each tested specimen (i.e. tensile strength, yield strength and young modulus) and even a statistic of all the tensile properties (i.e. average, standard deviation, and relative standard deviation) gathered for each build orientation of the tensile data.

Section 3 – The so-called standard points presented in Fig. 3, where do they come from? From the Fatigue results in the SN curve can one conclude an Endurance Limit?

It is reported the σ-N curve of standard Ti6Al4V in annealed condition [49]. In this review it was added another section (5 Discussion) where the results reported in section 4 are deeply discussed.  In this section  it is added the following consideration: “the σ-N curve of “Ti6Al4V machined” show a fatigue behavior comparable with that one of Ti6Al4V standard (in annealed condition) [49], even if the former seems don’t reach a fatigue limit.

Section 3 -  is called Results and Discussion, however the authors only present the results in a poor way, lacking discussion as to why the differences between AM and AM-machined components have different mechanical properties. How is the building orientation affecting the results?

The section 3 was split in two sections: section 3 Results and section 4 Discussion. Section 3 was divided into three sub-sections: 3.1 Tensile tests; 3.2 Fatigue tests; 3.3 Linear-elastic plane-strain fracture toughness KIC tests. In these subsection were added more informations on tests results:

3.1 (table 3 and table 4);

3.2 Fracture surfaces analysis;

3.3 A photo of the fracture surface showing no plastic deformation.

Reviewer 2 Report

Abstract

“Electron Beam Melting (EBM) is one of a few AM (Additive Manufacturing) technologies
capable of making full-density functional metallic parts realized from raw materials in the form of powders. EBM utilizes a high-energy electron beam, as a moving heat source, to melt and solidify, by rapid self-cooling, metal powder and produce parts in a layer-building fashion and freeform possibilities for designing especially for complex components, e.g., fine network structures, internal cavities and channels, which are difficult to make by conventional manufacturing means.”

Focus on what and how are the problems to be solved in this study in addition to the major findings.

In addition, be concise and brief.

There are no significant innovations in this study as most of the main contents, including experiment design and conclusions. So please compare it with former studies and focus on innovations.

Introduction

From the first line 27 to line 64

These words are more likely for technique report.

Please do a better job by review the most important studies in this field for literature review. There are tons of studies in EBM Ti64. Need give a summary of the former studies and then state what problems will be solved in this study.

For the general AM technical related things, just need one or two sentence giving a brief introduction by adding some references in case the readers want to know more.

Figure 2,

Please add standard deviation values.

Enlarge the font size in the images.

“It is well known from literature that bad fatigue performance is mainly due to the high surface”

References are needed.

“Linear-elastic plane-strain fracture toughness KIC tests, compliant to the ASTM E399, have been
 performed on n.6 compact CT samples shown in Fig. 4.”

General past tense would be better.

Same for other paragraphs.

Ti6Al4V standard (in annealed condition) [26] which is 74,6 [MPa√m]

Here, suggest using MPa*m-2

Author Response

There are no significant innovations in this study as most of the main contents, including experiment design and conclusions. So please compare it with former studies and focus on innovations.

A more detailed analysis of the literature was done on Ti6Al4V “produced by EBM”: microstructure and mechanical characterization. In particular for tensile, fatigue and fracture toughness properties of Ti6Al4V processed by EBM was analyzed and discussed the effect of the anisotropies and of the defects (porosity, lack of fusions, surface morphology and roughness). The analysis of the state of the art on Ti6Al4V produced by EBM gives the chance of opening a reflection on investigations of possible post-processings in order to improve mechanical performance of the material. At this aim this work try to give information on tensile, fatigue and fracture toughness behavior of Ti6Al4V produced by EBM. The scientific contribution of this work consists in the investigations on post processing, as well the machining which was analyzed in this work, and how it can lead to an improvement of the mechanical properties of the Ti6Al4V produced by EBM. This is very important since quality and reliability of the components are needed for aerospace applications, above all in aerospace field where it lacks a procedure for certifying AM components production. In fact the results and the analysis of the literature have pointed out that the mechanical performances of Ti6Al4V produced by EBM are influenced by many factors: orientations, post processings, typical EBM defects. The discussion afforded on the mechanical characterization and the possible post processing improving Ti6Al4V-EBM performances is a contribution to increase the comprehension of the EBM process  and its applications. In conclusion the main innovation of this work is the achievement of data generation that allows the development of aeronuatical principal structural element by using Additive Manufacturing technology. That comprises the new design approach and related features, characterization of the process in order to study the strategy and the possibility to obtain a future flight clearance and the identification of the best post-processing practice (if necessary).

Introduction

From the first line 27 to line 64

These words are more likely for technique report.

Please do a better job by review the most important studies in this field for literature review. There are tons of studies in EBM Ti64. Need give a summary of the former studies and then state what problems will be solved in this study.

Section 1 was completely reviewed. All the information concerning the project (lines27-64) was eliminated. A more detailed analysis of the literature was done on Ti64 “produced by EBM”: microstructure and mechanical characterization. In particular for tensile, fatigue and fracture toughness properties of Ti6Al4V processed by EBM was analyzed the effect of the anisotropies and of the defects (porosity, lack of fusions, surface morphology and roughness). The analysis of the state of the art on Ti6Al4V produced by EBM gives the chance of opening a reflection on investigations of possible post-processings  in order to improve mechanical performance of the material. At this aim this work try to give information on tensile, fatigue and fracture toughness behavior of Ti6Al4V produced by EBM. The post processings, as well the machining which was investigated in this work, can lead to an improvement of the mechanical properties of the Ti6Al4V produced by EBM. This very important since quality and reliability of the components are needed for aerospace applications, above all in aerospace field where it lacks a procedure for certifying AM components production.

For the general AM technical related things, just need one or two sentence giving a brief introduction by adding some references in case the readers want to know more.

In section 1  it was carried out a brief introduction on  the industrial needs which have pushed research into the development of new manufacturing processes as for example the Additive Manufacturing (AM) technology. The definition of AM was given citing the ASTM Standard F2729-12, and a historical view on the AM processes development from rapid prototyping to the digital manufacturing of metallic components named additive manufacturing was done. It was explored the EBM technology as the most promising AM technology for processing titanium alloys and in particular the Ti6Al4V alloy, as well as materials that require elevated process temperatures. Briefly the EBM advantages in processing the Ti6Al4V  were even mentioned.

Figure 2,

Please add standard deviation values.

Enlarge the font size in the images.

The font size of figure 2 was enlarged. As required two tables (tab.3 and tab.4) were added in which are shown the tensile properties values of each tested specimen (i.e. tensile strength, yield strength and young modulus) and even a statistics of all the tensile properties (i.e. average, standard deviation, and relative standard deviation) gathered for each build orientation of the tensile data.

“It is well known from literature that bad fatigue performance is mainly due to the high surface”

References are needed.

The reference was added. “It is well known from literature that bad fatigue performance is mainly due to the high surface roughness and/or high internal porosity [52].”

“Linear-elastic plane-strain fracture toughness KIC tests, compliant to the ASTM E399, have been
 performed on n.6 compact CT samples shown in Fig. 4.”

General past tense would be better.

done

Same for other paragraphs.

done

Ti6Al4V standard (in annealed condition) [26] which is 74,6 [MPa√m] should be written with exponential

done

Reviewer 3 Report

In this work, the authors characterize EBM-processed Ti6Al4V and report the results of the experiments. The work has some promise, but I have some concerns with publication of the study and paper as written, as described below. I recommend that the handling editor return the paper to the authors for a major revision to address the concerns before another round of reviews. The paper seems like a short conference paper or technical report as written and will need a major expansion (+10 pages or more) before it could be acceptable for publication in any archival journal - but it could be done if the authors are willing to put in the work. I do not think any additional experiments are needed, so most of the work will be on presentation and expansion of the paper. Please note that all comments are meant to help the authors and ensure the quality of the literature and are in no way pointed at the authors personally.  

Major comments

1. I do not see a significant contribution to aerospace engineering or aeronautics with this paper. It has possibility, but I do not see why it was submitted to this specific journal. There is some very minor mention of a standards framework for aerospace parts but this is not expanded or developed by the authors so this cannot be counted as a contribution. This paper either needs to extend that contribution or transfer the paper to a journal related to materials processing or general mechanics. 

2. There have been many similar studies completed on EBM Ti6Al4V. What is the novelty of the study? What new information does this paper provide that cannot be found by completing a careful literature review? This Ti alloy is by far the most common EBM material (and aerospace is not the most common application of EBM as it is used far more in medical) and there are dozens of papers characterizing it, many of them far more extensive and rigorous than this one. The literature review on this is poor and does not summarize previous results to highlight the contribution of the new ones presented here.  

3. The characterization studies are not extensive enough or deep enough for a full characterization paper (all of the cited ASTM standards require 5-10 replications - some of the tests have only 3). This is fine as long as the data is being used for something else, such as to build a framework or for extracting design knowledge. But if the authors are representing this as a characterization (i.e., development of a data catalog for the material), there needs to be more justification for the low number of tests and the almost complete lack of analysis of the results. 

4. There is essentially no deep analysis of the results of the experiments. For example, you got a fairly low K_IC (K_IC for cast or machined Ti-Al alloys is about 3x higher than this) and high StDev for the fatigue tests. What does this mean? What conclusions can you make about these results? Show the cracks - does the K_IC correlate very well to crack lengths and path non-straightness? I would expect to see at least several pages of discussion related to an experiment series like this, especially if the authors are trying to make a case for novelty in an area with many competing papers. 

5. The experimental section is not detailed enough for me to reproduce the experiments. Please expand this and give all the processing details.

6. I would like to see a lot more details about the fatigue and fracture tests. Where are the curves for the tests? At least give some example curves. Provide some pictures of the samples, especially the tested ones. I would like to see microscope images of the crack fronts as well (for reasons mentioned in other comments). 

7. Some of the discussion in the paper about the fatigue properties is attributed to surface roughness and porosity, but there is very little mention about the residual stresses, which likely have as much or more impact on this. 

8. In several places, the authors mention their SAT-AM project. Why is this mentioned in the text? It should be mentioned in the acknowledgements (since it seems this project provided funding) but this is not relevant information to anyone outside of the immediate research group of the authors. Present the results and make conclusions that are relevant to the readership of an international journal only. If the authors want to report on their project only, they should publish a technical report through their university and not submit the report as a research paper to an international journal. 

Specific comments

9. Lines 15-16: This paper makes no real mention about this certification process after the abstract. This is very interesting and would be a major contribution, but the authors mostly ignore it after this point. 

10. Lines 29: Again, the specific project the authors are working on is irrelevant information in this context. Provide context from the other published literature on the topic and make the paper stand on its own.

11. Lines 76-77: This objective is good, but it is not accomplished with this study. Providing some material data that already exists in the literature is not a major contribution in itself. The literature review is very poor for this paper, focusing on irrelevant information like the history of AM and the authors' project - it should focus on material properties, processing and testing parameters, etc related to the presented topic. 

12. Figure 3: In materials science and materials processing, the "S-N curve" is generally the signal-to-noise curve used to find statistical information for the dataset. It is stress-cycles curve in this study but this needs to be more clearly pointed out here as it may cause confusion to the reader. 

13. Section 3.3: You don't know this is LEFM until the results are analyzed. I expect it is indeed LEFM from the data but you also need to examine the crack tips and try to determine the size of the plane-strain field for the CT samples to make this claim. 

14. The paper seems to abruptly end after Figure 4. Are there more sections? The paper is nowhere close to complete. 

Author Response

Reviwers 3

In this work, the authors characterize EBM-processed Ti6Al4V and report the results of the experiments. The work has some promise, but I have some concerns with publication of the study and paper as written, as described below. I recommend that the handling editor return the paper to the authors for a major revision to address the concerns before another round of reviews. The paper seems like a short conference paper or technical report as written and will need a major expansion (+10 pages or more) before it could be acceptable for publication in any archival journal - but it could be done if the authors are willing to put in the work. I do not think any additional experiments are needed, so most of the work will be on presentation and expansion of the paper. Please note that all comments are meant to help the authors and ensure the quality of the literature and are in no way pointed at the authors personally.  

The paper was completely reviewed in its structure. A more detailed analysis of the literature was done. The section Materials and Methods was expanded with three subsections one for each type of  performed test. In these sub-sections it was explained the methods, the machines, the setups used for the testing and information on post-processing (machining). Moreover in these subsections were added several images of samples drawings and photos of the samples pre and after post processing. The section 3 was changed too. Three sub-sections (one for each mechanical test performed) were added. And section 4 was added in order to deeply afford the discussion of tests results from a scientific and speculative point of view.

Major comments

  1. I do not see a significant contribution to aerospace engineering or aeronautics with this paper. It has possibility, but I do not see why it was submitted to this specific journal. There is some very minor mention of a standards framework for aerospace parts but this is not expanded or developed by the authors so this cannot be counted as a contribution. This paper either needs to extend that contribution or transfer the paper to a journal related to materials processing or general mechanics. 

  In Section 1 it was added from line 87 to 134 a wide introduction on the certification issue.  The scientific contribution of this work to the aerospace engineering consists in the investigations on post processing, as well the machining which was herein analyzed and how it can lead to an improvement of the mechanical properties of the Ti6Al4V produced by EBM. This is very important since quality and reliability of the components are needed for aerospace applications, above all in aerospace field where it lacks a formal procedure for certifying AM components production. In fact the results and analysis of the litterature  have pointed out that the mechanical performances of Ti6Al4V produced by EBM are influenced by many factors: orientations, post processings, typical EBM defects. The discussion afforded on the mechanical characterization and the possible post processing (i.e. machining and marginally even HIP) improving Ti6Al4V (EBM) performances is a contribution to increase the comprehension of the EBM process. At the same time it provides useful information to designers in order to improve their confidence on components designing, and to technologists in order to choice better EBM manufacturing strategies. The final aim of the studies presented are not the (partial) mechanical characterization of specific Ti6AL4Vspecimens manufactured via EBM. The main innovation of this study is the achievement of data generation that allow the development of aeronatical  principal structural element by using Additive Manufacturing technology. That comprises the new design approach and related features, characterization of the process in order to study the strategy and the possibility to obtain a future flight clearance and the identification of the best post-processing practice (if necessary). The last two streams are included in this study.   

2. There have been many similar studies completed on EBM Ti6Al4V. What is the novelty of the study? What new information does this paper provide that cannot be found by completing a careful literature review? This Ti alloy is by far the most common EBM material (and aerospace is not the most common application of EBM as it is used far more in medical) and there are dozens of papers characterizing it, many of them far more extensive and rigorous than this one. The literature review on this is poor and does not summarize previous results to highlight the contribution of the new ones presented here.  

Literature review (Section 1) was completely reviewed. All the information concerning the project was eliminated. In this section from line  it was done a brief introduction on  the industrial needs which have pushed research into the development of new manufacturing processes as for example the Additive Manufacturing (AM) technology. The definition of AM was given citing the ASTM Standard F2729-12, and a historical view on the AM processes development from rapid prototyping to the digital manufacturing of metallic components named additive manufacturing was done. It was explored the EBM technology as the most promising AM technology for titanium alloys and in particular the Ti6Al4V alloy, as well as materials that require elevated process temperatures. Briefly the EBM advantages in processing the Ti6Al4V were even mentioned. A more detailed analysis of the literature was done on Ti64 “produced by EBM”: microstructure and mechanical characterization. In particular for tensile, fatigue and fracture toughness properties of Ti6Al4V processed by EBM was analyzed the effect of the anisotropies and of the defects (porosity, lack of fusions, surface morphology and roughness). The analysis of the state of the art  on Ti6Al4V produced by EBM gives the chance of opening a reflection on investigations of possible post-processings in order to improve mechanical performance of the material. At this aim this work try to give information on tensile, fatigue and fracture toughness behavior of Ti6Al4V produced by EBM. The post processings, as well the machining which was investigated in this work, can lead to an improvement of the mechanical properties of the Ti6Al4V produced by EBM. This very important since quality and reliability of the components are needed for aerospace applications, above all in aerospace field where it lacks a procedure for certifying AM components production. In fact the results and the analysis of the literature have pointed out that the mechanical performances of Ti6Al4V produced by EBM are influenced by many factors: orientations, post processings, typical EBM defects. The discussion afforded on the mechanical characterization and the possible post processing improving Ti6Al4V (EBM) performances is a contribution to increase the comprehension of the EBM process. At the same time it provides useful information to designers in order to improve their confidence on components designing, and to technologists in order to choice better EBM manufacturing strategies. In conclusion the main innovation of this work is the achievement of data generation that allows the development of aeronautical principal structural element by using Additive Manufacturing technology. That comprises the new design approach and related features, characterization of the process in order to study the strategy and the possibility to obtain a future flight clearance and the identification of the best post-processing practice (if necessary).

  1. The characterization studies are not extensive enough or deep enough for a full characterization paper (all of the cited ASTM standards require 5-10 replications - some of the tests have only 3). This is fine as long as the data is being used for something else, such as to build a framework or for extracting design knowledge. But if the authors are representing this as a characterization (i.e., development of a data catalog for the material), there needs to be more justification for the low number of tests and the almost complete lack of analysis of the results. 

In sub section 3.1Tensile tests, two tables (tab.3 and tab.4) were added in which are shown the tensile properties values of each tested specimen (i.e. tensile strength, yield strength and young modulus) and even a statistic of all the tensile properties (i.e. average, standard deviation, and relative standard deviation) gathered for each build orientation of the tensile data. The tensile test results have shown a very high repeatability as even confirmed by their small values of the standard deviations and of the relative standard deviations calculated in table 3 and tab. 4. For this reason it is possible to affirm that that tensile data resulting from this work can be considered statistically reliable even if three only specimens were tested.

  1. There is essentially no deep analysis of the results of the experiments. For example, you got a fairly low K_IC (K_IC for cast or machined Ti-Al alloys is about 3x higher than this) and high StDev for the fatigue tests. What does this mean? What conclusions can you make about these results? Show the cracks - does the K_IC correlate very well to crack lengths and path non-straightness? I would expect to see at least several pages of discussion related to an experiment series like this, especially if the authors are trying to make a case for novelty in an area with many competing papers. 

The section 3 was split in two sections: 3 Results and 4 Discussion .

Section 3 was divided into three sub-sections: 3.1 Tensile tests; 3.2 Fatigue tests; 3.3 Linear-elastic plane-strain fracture toughness KIC test. In these subsection were added more informations on tests results:

3.1 table 3 and table 4, where are reported tensile data and their statistics;

3.2 Fracture surfaces analysis (for fatigue tested specimens)

3.3 A photo of the fracture surface showing no plastic deformation.

In Section 4 it was done a discussion on tests results with the aim to give explanations of the tensile, fatigue and KIC results of the Ti6Al4V produced by EBM as function of typical α+β (titanium) microstructure (as mentioned even in the analysis of the literature – Section1)  and of the typical defects (as porosity, lack of fusions, roughness and surface morphology). The post processing as well machining was discussed and its beneficial effect on the improvement of the mechanical behavior was explained.

  1. The experimental section is not detailed enough for me to reproduce the experiments. Please expand this and give all the processing details

Section 2 was completely changed. This section was expanded with three subsections one for each type of test performed. In these sub-sections it was explained the methods, the machines, the setups used for the testing and information on post-processing (machining). Moreover in these subsections were added several images of samples drawing and photos of the samples pre and after post processing.

.

  1. I would like to see a lot more details about the fatigue and fracture tests. Where are the curves for the tests? At least give some example curves. Provide some pictures of the samples, especially the tested ones. I would like to see microscope images of the crack fronts as well (for reasons mentioned in other comments). 

In the sub-section 3.2 the σ-N curves were provided (fig.9) and analysis of the fracture surface  for both as built and machined specimens too (fig.10 and fig.11)

In the sub-section 3.3 typical graph force – crack mouth displacement was reported (fig.12) and even a photo of the fracture surface showing no plastic deformation (fig 13)

  1. Some of the discussion in the paper about the fatigue properties is attributed to surface roughness and porosity, but there is very little mention about the residual stresses, which likely have as much or more impact on this. 

In the section 4 (Discussion) it was added a mention about the beneficial effect of the residual stresses induced by machining on the surfaces of the specimens. It was cited reference 53 in which it was highlighted the retarding effect of these residual stress on the crack nucleation.

  1. In several places, the authors mention their SAT-AM project. Why is this mentioned in the text? It should be mentioned in the acknowledgements (since it seems this project provided funding) but this is not relevant information to anyone outside of the immediate research group of the authors. Present the results and make conclusions that are relevant to the readership of an international journal only. If the authors want to report on their project only, they should publish a technical report through their university and not submit the report as a research paper to an international journal. 

 In the introduction (section 1) the information related to the SAT AM project has been streamlined and related directly to the present study aims.  

Specific comments

  1. Lines 15-16: This paper makes no real mention about this certification process after the abstract. This is very interesting and would be a major contribution, but the authors mostly ignore it after this point. 

In Section 1 it was added from line 88 -134 a wide introduction on the certification issue.

  1. Lines 29: Again, the specific project the authors are working on is irrelevant information in this context. Provide context from the other published literature on the topic and make the paper stand on its own.

Section 1 was completely reviewed. All the information concerning the project was eliminated.

  1. Lines 76-77: This objective is good, but it is not accomplished with this study. Providing some material data that already exists in the literature is not a major contribution in itself. The literature review is very poor for this paper, focusing on irrelevant information like the history of AM and the authors' project - it should focus on material properties, processing and testing parameters, etc related to the presented topic. 

Section 1 was completely reviewed. All the information concerning the project  was eliminated and reduced the history of AM technology. A more detailed analysis of the literature was done on Ti6Al4V “produced by EBM”: microstructure and mechanical characterization. In particular for tensile, fatigue and fracture toughness properties of Ti6Al4V processed by EBM was analyzed and discussed the effect of the anisotropies and of the defects (porosity, lack of fusions, surface morphology and roughness). The analysis of the state of the art on Ti6Al4V produced by EBM gives the chance of opening a reflection on investigations of possible post-processings in order to improve mechanical performance of the material. At this aim this work try to give information on tensile, fatigue and fracture toughness behavior of Ti6Al4V produced by EBM. The post processings, as well the machining which was investigated in this work, can lead to an improvement of the mechanical properties of the Ti6Al4V produced by EBM. This very important since quality and reliability of the components are needed for aerospace applications, above all in aerospace field where it lacks a procedure for certifying AM components production.

  1. Figure 3: In materials science and materials processing, the "S-N curve" is generally the signal-to-noise curve used to find statistical information for the dataset. It is stress-cycles curve in this study but this needs to be more clearly pointed out here as it may cause confusion to the reader.

The S-N curves were substituted with σ-N curves in order to avoid misunderstandings

  1. Section 3.3: You don't know this is LEFM until the results are analyzed. I expect it is indeed LEFM from the data but you also need to examine the crack tips and try to determine the size of the plane-strain field for the CT samples to make this claim. 

In section 3.3 it was added an optical analysis of the fracture surfaces (fig.13) highlighting that no plastic deformation were observed showing an evident brittle fracture mode. For this reason the Linear-elastic plane-strain fracture toughness tests was guaranteed to be resulted in a KIC value measured under plane-strain conditions.

  1. The paper seems to abruptly end after Figure 4. Are there more sections? The paper is nowhere close to complete. 

Section 4 was added, where it was carried out a discussion on tests results with the aim to give explanations of the mechanical behavior of the Ti6Al4V produced by EBM as function of typical α+β microstructure (as mentioned even in the analysis of the literature – Section1)  and of the typical defects (as porosity, lack of fusions, roughness and surface morphology). The post processing, as well machining, was discussed and its beneficial effect on the improvement of the mechanical behavior was explained.

Round 2

Reviewer 1 Report

The manuscript has a significant improvement. I recommend acceptance upon some minor issues listed here:

  • L96 – there is an error with one of the references.
  • Figure 2 seems to need references or CopyRight permissions. 

Author Response

  • L96 – there is an error with one of the references.

Reply: It was corrected the reference [35] (L 106)

  • Figure 2 seems to need references or CopyRight permissions. 

Reply: This is a CIRA’s drawing. CopyRight permissions are not needed. (L 201)

Reviewer 2 Report

Title

It is too big for this study. As it aims to

investigate the effect of post processing as machining on the mechanical behavior.

Please change the title.

Make sure the title can reflect the content of the study.

In addition, EBM is not allowed to be used in the title. Full name words should be used.

.

EBM is not the standard terminology. According to ASTM,

electron beam powder bed fusion additive manufacturing

is the more professional name. After stating this in the manuscript, Electron Beam Melting would be good to use in the manuscript.

Abstract

Too long. Be brief and concise.

Focus on the major findings in this study.

Introduction

The 1st paragraph should be divided into multiple paragraphs.

Too long to be understandable.

Same for paragraph 2.

For the literature review, please also include former important studies, such as, Comparison of the microstructures and mechanical properties of Ti–6Al–4V fabricated by selective laser melting and electron beam melting, and EBSD study of beam speed effects on Ti-6Al-4V alloy by powder bed electron beam additive manufacturing

96 additive manufactured parts [Error! Reference source not found.]. For this reason more studies on

Modify.

“These projects are committed to speed up the entering of AM technology in
128 the sector and to try to collect the technical and scientific data required by Authorities to aid the
129 comprehension of AM process (dependent on specific manufacturer practice), its limits and its level”

At the end of the introduction,

Please stating clearly what problems will be solved and what are the innovations in this study. As stated below,

- “As built” directly EBM manufactured in the dogbone shape suitable to be tested
- “Machined” obtained by machining cylindrical bars produced by EBM This is not new,

Has anyone done this comparison before?

A test matrix showing samples information as type manufacturing condition, build orientation and
164 the related standard used is summarized in Table 2.

How these parameters were determined? Please state clearly why using these parameters.

The
149 powder nominal chemical composition is summarized in Table 1.

How these values were obtained? Please add details

Table 1. Nominal chemical composition (wt.%) of the Ti6Al4V pre-alloyed powder used in EBM

Please use a period, not a comma in the numbers. Same for below.

Figure 2. Drawing of tensile specimens (the dimensions are referred to mm) 

Figure 4. Drawing of fatigue specimens (the dimensions are referred to mm)

Why repeated?

160 two type of manufacturing conditions:
161 - “As built” directly EBM manufactured in the dogbone shape suitable to be tested
162 - “Machined” obtained by machining cylindrical bars produced by EBM

176 The specimens were manufactured in two manufacturing conditions (fig.3):
177 - “As built” directly EBM manufactured in the dog-bone shape suitable to be tested
178 - “Machined” obtained by machining cylindrical bars produced by EBM

182 For each manufacturing conditions (i.e. “as built” and “machined”), three sets of specimens were
183 produced:
184 - n.3 tensile samples with 0° build orientation with respect to the start plate (x-y plane);
185 - n.3 tensile samples with 45° build orientation with respect to the start plate (x-y plane);
186 - n.3 tensile samples with 90° build orientation with respect to the start plate (x-y plane).

Try to explain it using a diagram.

Same for

190 2.2. Fatigue Tests
191 Two sets of Ti6Al4V specimens were manufactured with 90° build orientation with respect to the
192 start plate (x-y plane) and tested to fatigue. In particular were EBM-manufactured:

More words do not mean clear. Try to express the idea in a different way.

Figure 8. a) Young modulus values; b) Yield strength values; c) Tensile strength values

How about standard deviations?

Table 3. Test data of as built specimens and the related statistics

Table 4. Test data of as built specimens and the related statistics

In addition to the table, suggest also plots them for easier comparison.

  1. Conclusions

Suggest use bullet points listing the major findings one by one.

Author Response

Question 1) Title

It is too big for this study. As it aims to

investigate the effect of post processing as machining on the mechanical behavior.

Please change the title.

Make sure the title can reflect the content of the study.

Answer 1) The effect of post processing on the mechanical behavior of Ti6Al4V manufactured by electron beam powder bed fusion for General Aviation primary structural applications (L1).

In addition, EBM is not allowed to be used in the title. Full name words should be used.

Question 2) EBM is not the standard terminology. According to ASTM,

electron beam powder bed fusion additive manufacturing

is the more professional name. After stating this in the manuscript, Electron Beam Melting would be good to use in the manuscript.

Answer 2) The extensive name of electron beam powder bed fusion additive manufacturing was added in the  (L 1) as suggested in according to the terminology of ASTM . Then electron beam melting was mentioned in full version and as acronym (L 45)

Question 3) Abstract

Too long. Be brief and concise.

Focus on the major findings in this study.

Answer 3) Abstract was reduced to less than 200 words (L12-25)

Question 4) Introduction

The 1st paragraph should be divided into multiple paragraphs.

Too long to be understandable.

Answer  4) The 1st paragraph was dived into two paragraphs:

1 Introduction (L29)

2 Major issues of EBM use for Aircraft application (L94)

Question 5) Same for paragraph 2.

Answer 5) The 2ndparagraph was dived into two paragraphs:

2 Materials (L156)

3 Methods (L181)

Question 6) For the literature review, please also include former important studies, such as, Comparison of the microstructures and mechanical properties of Ti–6Al–4V fabricated by selective laser melting and electron beam melting, and EBSD study of beam speed effects on Ti-6Al-4V alloy by powder bed electron beam additive manufacturing

Answer 6) In the paragraph 1 it was added some studies on microstructures and mechanical properties of Ti–6Al–4V fabricated by selective laser melting and electron beam melting:

 “While the EBM microstructure shows the a-phase grain structure and b-boundary areas, the SLM microstructure consists a’-martensite platelets. The EBM microstructure shows columnar grain boundaries generally parallel to the build direction. The a’-martensite forms in preference to the acicular a-phase because of the more rapid solidification in SLM processing in contrast to EBM processing [22-24]” (L59-63)

“The tensile strength of the SLM specimens were similar to those of the EBM specimens even if the latter were slightly lower than the former ones” [13, 22] (L67-68)

In the same paragraph 1 it was added an EBSD study of beam speed effects on Ti-6Al-4V alloy by powder bed electron beam additive manufacturing

“Investigation on process parameters aimed at improving Ti6Al4V microstructure produced by EBM were carried out in [25]. The increase of electron beam scanning speed did not show significant effects on the orientation of the grains in the z-plane, on the contrary in the y-plane significant changes in the preferred orientation were appreciated. The microstructure evolution due to the increase of electron beam scanning speed significantly reduced anisotropy in such properties as hardness and elastic modulus.”(L68-74)

Question 7) 96 additive manufactured parts [Error! Reference source not found.]. For this reason more studies on

Answer 7) It was corrected with the right reference [35] (L106)

Question 8) Modify.

“These projects are committed to speed up the entering of AM technology in
128 the sector and to try to collect the technical and scientific data required by Authorities to aid the
129 comprehension of AM process (dependent on specific manufacturer practice), its limits and its level”

Answer 8) The sentence was modified as it follows:

“These projects are aimed to promote and grow up AM technology in the aviation sector providing a set of data on mechanical and microstructural properties required by Authorities to aid the comprehension of AM process (dependent on specific manufacturer practice), Its limits and its level of reliability.” (L137-140)

Question 9) At the end of the introduction,

Please stating clearly what problems will be solved and what are the innovations in this study. As stated below,

- “As built” directly EBM manufactured in the dogbone shape suitable to be tested
- “Machined” obtained by machining cylindrical bars produced by EBM This is not new,

Has anyone done this comparison before? 

Answer  9) Many other studies were carried out on Ti6Al4V produced by EBM in as built and machined conditions. Anyway in paragraph 2 it was tryed to explain the novelty of this work

9) “Even if many other works were carried out on the mechanical characterization of Ti6Al4V produced by EBM with and without post processing the current study differs from the other ones for its specific objective. It wants to give additional knowledge to General Aviation (GA) Aircraft development current practice in case of use of a new (with respect GA specific sector) manufacturing process. More in details, the authors have identified the basic data to support the GA structural substation and the design loop (i.e. mechanical characteristics and their statistics variability). In addition, clear indication on how the post processing reworks can improve the structural performances of any Primary Structure Element (PSE) has been given by the authors. Although, the rework can affect the total recurrent cost and the delivery time of the PSE, contributing to decrease the AM specific technology appeal. It is deemed crucial to have quantified the benefit of rework in terms of mechanical proprieties improvement giving to the designers the suitable quantitative data to drive their decision. All those considerations can be considered as a very first application for a GA PSE.” (L 143- L 155)

Question 10) A test matrix showing samples information as type manufacturing condition, build orientation and 164 the related standard used is summarized in Table 2.

How these parameters were determined? Please state clearly why using these parameters.

Answer 10) In the case of tensile behavior, considering the slight anisotropy of Ti6Al4V processed by EBM  (even if for the cited study [25] (L66) the tensile properties  anisotropy are considered not significant, except for the elongation at break), it was decided to test all three directions 0°, 45°, 90°.

For fatigue and fracture toughness behavior, for sake of brevity, it was chosen to test only the build orientation associated to the weakest perfomace. As stated in the article:

“Contrary to what noted for the elongation at break, the anisotropy of fatigue behavior consists in a significant lower strength for samples built in the vertical orientation [28, 30]. Even fracture toughness shows a significant anisotropy due to the different propagation of the crack in samples built in horizontal and vertical orientation. In [31-33] it is highlighted that the samples built in vertical orientation and with horizontal notch (i.e. parallel to the building layer) show a significant worse fracture toughness value (KIC) than those ones fabricated in horizontal orientations “ (L82 – L 88)

Question 11) The 149 powder nominal chemical composition is summarized in Table 1

How these values were obtained? Please add details

Answer 11) It was added more information on powder nominal composition as it follows:

“The whole Ti6Al4V powder characterization was provided by the supplier (ARCAM company) with a certificate of analysis. Such a document contained, for the supplied batch of raw material, the results of the powder characterization in according to the above mentioned ASTM tests; i.e. the values of the powder flow rate, the apparent density, the particle distribution size and the chemical composition.” (L169-173)

Question 12) Table 1. Nominal chemical composition (wt.%) of the Ti6Al4V pre-alloyed powder used in EBM

Please use a period, not a comma in the numbers. Same for below.

Figure 2. Drawing of tensile specimens (the dimensions are referred to mm) 

Figure 4. Drawing of fatigue specimens (the dimensions are referred to mm)

Answer  12) Modified as requested: fig.2 (L 201)  and fig.4 (L 239)

Question 13) Why repeated?

160 two type of manufacturing conditions:
161 - “As built” directly EBM manufactured in the dogbone shape suitable to be tested
162 - “Machined” obtained by machining cylindrical bars produced by EBM

176 The specimens were manufactured in two manufacturing conditions (fig.3):
177 - “As built” directly EBM manufactured in the dog-bone shape suitable to be tested
178 - “Machined” obtained by machining cylindrical bars produced by EBM

Answer 13) it was corrected with the following sentence

“The specimens were manufactured in two manufacturing conditions (“as built” and “machined”  as even shown in Figure.3.)” (L 205-L 206)

Question 14) 182 For each manufacturing conditions (i.e. “as built” and “machined”), three sets of specimens were
183 produced:
184 - n.3 tensile samples with 0° build orientation with respect to the start plate (x-y plane);
185 - n.3 tensile samples with 45° build orientation with respect to the start plate (x-y plane);
186 - n.3 tensile samples with 90° build orientation with respect to the start plate (x-y plane).

Try to explain it using a diagram.

Question 14) It was added a very explicative image (Figure 4) of the samples build orientations respect with the start plate (x-y plane) (L 220)

Question 15) Same for

190 2.2. Fatigue Tests
191 Two sets of Ti6Al4V specimens were manufactured with 90° build orientation with respect to the
192 start plate (x-y plane) and tested to fatigue. In particular were EBM-manufactured:

More words do not mean clear. Try to express the idea in a different way.

Question 15) The sentence was modified as follows

“Two sets of Ti6Al4V specimens were manufactured and tested to fatigue. Both the sets of specimens were produced with 90° build orientation with respect to the start plate (x-y plane), that is in the vertical direction. In particular were EBM-manufactured:” (L 225 – L226)

Question 16) Figure 8. a) Young modulus values; b) Yield strength values; c) Tensile strength values

How about standard deviations?

Table 3. Test data of as built specimens and the related statistics

Table 4. Test data of as built specimens and the related statistics

In addition to the table, suggest also plots them for easier comparison.

Question 16) The cited figure 8 was substituted with Figure 9. This figure shows averages and standard deviations of tensile tests results on behalf of the previous graphs (fig.8) where the tensile data were represented by histograms showing only their avarages. (L 290)

Question 17) Conclusions

Suggest use bullet points listing the major findings one by one.

Question 17) Conclusions were modified using bullet points as follows

“Ti6Al4V EBM processed was tested in two different manufacturing conditions “as built” and “machined” (i.e. post-processed by machining).

The results can be summarized:

  • the tensile tests have shown high mechanical performance of Ti6Al4V EBM-processed, particularly the specimens obtained by machining bars produced by EBM have shown tensile results slightly better than standard Ti6Al4V in annealed condition [45];
  • the fatigue performance of Ti6Al4V processed by EBM are generally lower than Ti6Al4V standard (in annealed condition) [45], highlighting, on the other side, that Ti6Al4V produced by EBM in as built condition show the worst fatigue behavior;
  • the stress-cycles curve of Ti6Al4V obtained by machining cylindrical bars produced by EBM shows a comparable, even if slightly lower, behavior if compared with Ti6Al4V standard (in annealed condition)[45] ;
  • the KIC value of Ti6Al4V produced by EBM is considerably worse than Ti6Al4V standard (in annealed condition) [45.” (L 431 –L 444)

Reviewer 3 Report

The authors have done a significant revision of this paper and it is definitely improved in terms of style and presentation. However, I do not think the authors have yet done enough to establish the novelty of the method presented and to establish a stronger connection to the aerospace engineering field. I would also like all of the references to a particular research project or country removed from the abstract and front material of the paper. Just present the facts in a context useful for the whole field and discuss the research project as one possible application at the end of the paper. 

If the authors can further improve these areas in a minor revision, I would recommend the paper for acceptance pending opinions from the other reviewers. Overall I still think this is a fairly weak/incremental paper in a field with many excellent previous works but there is enough new information presented to justify archiving it in a journal without a limit on the number of published articles per year. When I define it as a weak paper, this is not a criticism of the experimental approach and the presentation, as these are sound; my issue is with the context of the paper and the small novel contribution presented. I would ask the authors in the future to please plan their research projects based on the many open problems in AM and aerospace engineering instead of effectively replicating existing work with a small new context. There are many open problems related to aerospace systems, design, standards and characterization methods, and polymer AM for which high-impact papers can be written at a much lower cost and effort than the work presented in this paper. 

Minor comments: The abstract is far too long (it should be 200 words) and there are reference errors in the text

Author Response

1) The authors have done a significant revision of this paper and it is definitely improved in terms of style and presentation. However, I do not think the authors have yet done enough to establish the novelty of the method presented and to establish a stronger connection to the aerospace engineering field. I would also like all of the references to a particular research project or country removed from the abstract and front material of the paper. Just present the facts in a context useful for the whole field and discuss the research project as one possible application at the end of the paper.

Reply: 1) The abstract was widely reviewed removing all the references to the project in the frame of which this study was born.

Even if we are aware that many other studies were carried out on mechanical testing of Ti6Al4V manuctured via EBM in as bult and machined conditions, in paragraph 2 it was tryed to explain the novelty of this work. This explanation is reproted below:

“Even if many other works were carried out on the mechanical characterization of Ti6Al4V produced by EBM with and without post processing the current study differs from the other ones for its specific objective. It wants to give additional knowledge to General Aviation (GA) Aircraft development current practice in case of use of a new (with respect GA specific sector) manufacturing process. More in details, the authors have identified the basic data to support the GA structural substation and the design loop (i.e. mechanical characteristics and their statistics variability). In addition, clear indication on how the post processing reworks can improve the structural performances of any Primary Structure Element (PSE) has been given by the authors. Although, the rework can affect the total recurrent cost and the delivery time of the PSE, contributing to decrease the AM specific technology appeal. It is deemed crucial to have quantified the benefit of rework in terms of mechanical proprieties improvement giving to the designers the suitable quantitative data to drive their decision. All those considerations can be considered as a very first application for a GA PSE.” (L 143- L 155)

2)If the authors can further improve these areas in a minor revision, I would recommend the paper for acceptance pending opinions from the other reviewers. Overall I still think this is a fairly weak/incremental paper in a field with many excellent previous works but there is enough new information presented to justify archiving it in a journal without a limit on the number of published articles per year. When I define it as a weak paper, this is not a criticism of the experimental approach and the presentation, as these are sound; my issue is with the context of the paper and the small novel contribution presented. I would ask the authors in the future to please plan their research projects based on the many open problems in AM and aerospace engineering instead of effectively replicating existing work with a small new context. There are many open problems related to aerospace systems, design, standards and characterization methods, and polymer AM for which high-impact papers can be written at a much lower cost and effort than the work presented in this paper.

Reply: 2) We are aware that many other issues need to be investigated on AM above when this innovative manufacturing technology have to deal with aerospace engineering applications. By the fact we are orienting our future research activities to the investigation of how the use of Ti6Al4V recycled powder influences the chemical and physical features of Ti6Al4V processed by EBM.

3) Minor comments: The abstract is far too long (it should be 200 words) and there are reference errors in the text

Reply: 3) The abstract was widely reviewed and reduced to less than 200 words. (L12-25)

Round 3

Reviewer 2 Report

NA